# Functional Constipation and Anorexia in Community-Dwelling Older Adults: Korean Frailty and Aging Cohort Study (KFACS)

**DOI:** 10.3390/ijerph18115754

**Published:** 2021-05-27

**Authors:** Eunjin Jeong, Jung A Kim, Byung Sung Kim, Chang Kyun Lee, Miji Kim, Chang Won Won

**Affiliations:** 1Department of Family Medicine, College of Medicine, Kyung Hee University, Kyung Hee University Medical Center, Seoul 02447, Korea; ejjeong312@gmail.com (E.J.); kimjunga111@gmail.com (J.A.K.); bskim7@khmc.or.kr (B.S.K.); 2Center for Crohn’s and Colitis, Department of Gastroenterology, Kyung Hee University College of Medicine, Seoul 02447, Korea; changkyun.lee@khu.ac.kr; 3Department of Biomedical Science and Technology, College of Medicine/East-West Medical Research Institute, Kyung Hee University, Seoul 02447, Korea; mijiak@khu.ac.kr; 4Elderly Frailty Research Center, Department of Family Medicine, College of Medicine, Kyung Hee University, Seoul 02447, Korea

**Keywords:** anorexia, functional constipation, older adults, aging

## Abstract

Anorexia is a relevant geriatric syndrome because it accounts for most malnutrition in older adults. Constipation has been suggested as a risk factor for anorexia. This study aimed to examine the association between anorexia and functional constipation in community-dwelling older adults. Data on 899 subjects aged 72–86 years were obtained from a follow-up survey of the Korean Frailty and Aging Cohort Study in 2018. Anorexia was assessed using the Simplified Nutritional Appetite Questionnaire (SNAQ), while functional constipation was diagnosed based on Rome IV criteria. Anorexia and functional constipation were present in 30.9% and 19.6% of the participants, respectively. Age, female sex, chewing problems, malnutrition, polypharmacy, low Mini-Mental Status Examination (MMSE) score, depressed mood, low serum albumin, and functional constipation were associated with anorexia in the univariate analysis. In the multivariate logistic regression, functional constipation was associated with anorexia (OR 1.478, 95% CI 1.038–2.104) after adjusting for age, female sex, and MMSE score. However, after further adjusting for depressed mood (OR 2.568) and chewing problems (OR 2.196), the relationship was no longer significant. This study showed that functional constipation is associated with anorexia in community-dwelling older adults, but this association is confounded by depressed mood and chewing problems.

## 1. Introduction

Anorexia due to aging can be defined as the loss of appetite and decreased food intake in the elderly, and may account for the largest portion of the cause of malnutrition in older adults [1]. It is the most common dietary change in the elderly, experienced by up to 30% of community-dwelling older adults [2]. Decreased dietary intake and physical exercise can result in a decline in muscle mass in the elderly, which can worsen anorexia by promoting physiological changes and aggravating pathologic and socio-functional factors, resulting in a vicious cycle [1,3,4].

Anorexia of aging can be initiated by physiological, pathological, and social changes due to aging [5]. Identified risk factors in the literature include oral and dental diseases, chewing and swallowing problems, pharmacological therapies such as proton pump inhibitors (PPIs) and opioids, cognitive impairment, depression, and constipation [6]. Several studies have used a multifactorial approach to determine the impact of reversible risk factors on and their contributions to the etiology of anorexia of aging to suggest optimal treatments [2,7,8].

The recognition of risk factors could be useful in preventing malnutrition through assessments that can recognize the possible causes of anorexia. Constipation, one of the most frequently experienced chronic gastrointestinal disorders in adults, may be an important treatable cause of anorexia in the elderly [9]. The consequences of constipation, which is more common in the elderly, can be serious [10]. Constipation may compromise social functioning and the ability to perform activities of daily living [11,12]. Less acutely, constipation leading to fecal impaction can present with anorexia, nausea, and pain associated with functional decline [13]. Quality of life also appears to be lower in older people with constipation than in those without constipation [14].

Previous studies on the relationship between anorexia and constipation have shown conflicting results. Landi et al. stated that constipation is an important risk factor for anorexia in older nursing home residents [7]. In a study conducted in acute ward and rehabilitation settings, Donini et al. showed that patients with anorexia suffered more from constipation [8]. In contrast, Ilhan et al. suggested that constipation is not significantly associated with poor appetite in the analysis of patients admitted to acute geriatric units [2]. However, there have been no studies on the association between constipation and anorexia in community-dwelling older adults. Moreover, all previous studies investigated constipation through simple subjective responses from patients without using a standardized questionnaire.

Therefore, we investigated the relationship between functional constipation according to the Rome IV criteria and anorexia of aging based on the total Simplified Nutritional Appetite Questionnaire (SNAQ) score in community-dwelling older adults using the database of the Korean Frailty and Aging Cohort Study (KFACS).

## 2. Materials and Methods

### 2.1. Study Population

The KFACS is a multicenter longitudinal study with a baseline survey conducted from May 2016 to November 2017 to identify factors that contribute to aging in community-dwelling individuals aged 70–84 years. Each center recruited participants using quota sampling stratified by age (70–74, 75–79, and 80–84 years with a ratio of 6:5:4) and sex (male and female, 1:1). All participants were ambulatory, with or without the use of walking aids. Follow-up was conducted at two-year intervals.

The questionnaire for evaluating functional constipation was first included in a follow-up survey conducted in 2018. Out of 1559 subjects aged 70–84 years enrolled in the baseline survey in 2016, 1279 participants revisited the center after two years for a follow-up survey. As shown in Figure 1, this cross-sectional study included 899 subjects, excluding those without available prescription information (*n* = 382) and those who received opioid prescriptions (*n* = 50).

The KFACS protocol was approved by the Institutional Review Board (IRB) of the Clinical Research Ethics Committee of the Kyung Hee University Medical Center, and all participants provided written informed consent (IRB number: 2015-12-103) [15]. Data anonymization was performed to protect the privacy of participants. This study was conducted according to the consensus ethical principles derived from guidelines, including the Declaration of Helsinki.

#### 2.1.1. Inclusion Criteria

The research subjects were required to meet all the following criteria:A person older than 70 years but younger than 84 years, currently living in the community, not planning to move within 2 years, and without dementia;A person who provides informed consent to participate.

#### 2.1.2. Exclusion Criteria

Research subjects who met any of the following conditions could not participate in this study.

A person with difficulty giving their opinion due to dementia;A person who is considered unable to comply with the study requirements or is deemed inappropriate based on an evaluation by the researcher.

### 2.2. Assessment of Anorexia of Aging

The SNAQ is a four-item single-domain questionnaire. Responses were made using a five-point, verbally labeled Likert-type scale (points):My appetite is (generally): very poor (1), poor (2), average (3), good (4), very good (5).When I eat (satiety state): I feel full after eating only a few mouthfuls (1), I feel full after eating about a third of a meal (2), I feel full after eating over half a meal (3), I feel full after eating most of the meal (4), I hardly ever feel full (5).Food tastes: very bad (1), bad (2), average (3), good (4), very good (5).Normally I eat (daily frequency of meal): less than one meal a day (1), one meal a day (2), two meals a day (3), three meals a day (4), more than three meals a day (5).

The total SNAQ score is the sum of each item score; lower scores indicate more appetite deterioration [16]. Possible scores range form 4 (worst) to 20 (best); a score of 13 and lower was considered to indicate anorexia of aging.

### 2.3. Assessment of Functional Constipation

Functional constipation is a functional bowel disorder presenting with obvious low-frequency or incomplete defecation. The presence of functional constipation was assessed based on the Rome IV criteria for functional constipation [17], as follows.
Must include two or more of the following:Straining during more than one-fourth (25%) of defecations;Lumpy or hard stools (Bristol stool scale 1–2) in more than one-fourth (25%) of defecations;Sensation of incomplete evacuation in more than one-fourth (25%) of defecations;Sensation of anorectal obstruction/blockage in more than one-fourth (25%) of defecations;Manual maneuvers to facilitate more than one-fourth (25%) of defecations (e.g., digital evacuation, support of the pelvic floor);Fewer than three spontaneous bowel movements per week;Loose stools are rarely present without the use of laxatives;Insufficient criteria for irritable bowel syndrome.

### 2.4. Other Measurements

Demographic information, including age, sex, residential environment (living alone or not), was investigated through a face-to-face interview. Alcohol and smoking habits were reported as health behavior. Malnutrition was defined as a Mini Nutritional Assessment (MNA) score ≤ 11 [18]. Physical activity was assessed using the metabolic equivalent of task (MET) minutes per week, calculated on the basis of the International Physical Activity Questionnaire in a general Korean population-based survey of older adults [15]. Weight changes within the past year, body mass index (BMI) calculated as weight divided by height squared, and the presence of polypharmacy (having five or more kinds of drugs per day) were investigated.

Chronic disease states (hypertension, type 2 diabetes mellitus, dyslipidemia, depression, dementia, osteoporosis, cerebrovascular disease, cardiovascular disease, kidney disease, liver disease, thyroid disease, and malignancy), problems in chewing function, Geriatric Depression Scale (GDS), cognitive functions using Mini-Mental State Examination (MMSE), Frontal Assessment Battery (FAB), and trail making test were assessed for analysis. Laboratory findings associated with nutritional status were also assessed, including total protein level (g/L), serum albumin level (g/L), high-sensitivity C-reactive protein (mg/L), and serum creatine kinase (IU/L). Blood samples after 8 h fast were taken at around 08:00 to ensure the reliability of the hormone test.

The use of the following medications affecting constipation was assessed: recent use of PPIs, H2 receptor blockers, other anti-acids, non-steroid anti-inflammatory drugs (NSAIDs), antihypertensive medications (calcium channel blockers (CCBs) and diuretics), iron supplements, calcium supplements, anti-Parkinsonian medications (L-dopa and dopamine agonists), anticholinergics, antipsychotics, tricyclic antidepressants (TCAs), serotonin selective reuptake inhibitors (SSRIs), serotonin norepinephrine reuptake inhibitors (SNRIs), zolpidem, and benzodiazepines [19,20].

### 2.5. Statistical Analysis

The data are presented as mean ± standard deviation (SD) or as percentages. Continuous variables were compared using the independent *t*-test and categorical variables using the chi-squared test and Fisher’s exact test. To identify common risk factors for anorexia of aging and functional constipation, the study classified the subjects into groups with anorexia of aging and normal appetite or by groups with functional constipation and normal defecation. Logistic regression analysis of anorexia of aging as an independent variable was performed. Multivariate analysis was performed for risk factors that significantly contributed to anorexia of aging and functional constipation. Logistic regression models were constructed to evaluate the association between functional constipation and anorexia due to aging based on confounding variables. Models were adjusted for age, female gender, polypharmacy, MMSE, depressive mood, and chewing problems. All statistical analyses were conducted using IBM SPSS Statistics for Windows (version 19.0. Armonk, NY, USA). Statistical significance was set at *p* < 0.05.

## 3. Results

### 3.1. Basic Features of Subjects with Anorexia

The basic characteristics of the subjects are shown in Table 1. Of the 899 participants, 30.9% (*n* = 278) and 19.6% (*n* = 176) were classified as having anorexia and functional constipation, respectively. The mean age of the anorexia group (78.7 ± 4.1 years) was higher than that of the non-anorexia group. More females (63.7%) than males had anorexia due to aging. The anorexia group was more likely to have chewing problems (53.6%), a higher rate of current smoking (18.7%), polypharmacy (46.8%), malnutrition (MNA screening score ≤ 11) (31.5%), and functional constipation (25.5%).

Although the mean MMSE score was within the normal range in both groups, the score was significantly lower in the anorexia group. Mood status assessed by the GDS in the anorexia group also showed significantly more depressionthan in the non-anorexia group. In laboratory findings, the mean albumin levels were significantly lower in the anorexia group. No difference was found in high-sensitivity C-reactive protein levels.

### 3.2. Logistic Regression of Variables Affecting Anorexia

As shown in Table 2, multivariate binary logistic regression models were constructed. The models were designed to evaluate the association between functional constipation and anorexia of aging after adjusting for confounding variables. The confounding variables were selected from the statistically significant variables for anorexia (Table 1). Malnutrition and lower levels of serum albumin were excluded from the multivariate analysis because they are consequences of anorexia of aging rather than the causes. Finally, the logistic regression model included age, female sex, chewing problems, polypharmacy, MMSE score, and depressed mood by GDS as confounding variables.

In the multivariate logistic regression, functional constipation was associated with anorexia (odds ratio (OR) 1.478, 95% confidence interval (CI) 1.038–2.104) after adjusting for age, female sex, and MMSE score. However, after further adjusting for depressed mood (OR 2.568, 95% CI 1.801–3.661) and chewing problems (OR 2.196, 95% CI 1.612–2.992), the relationship was no longer significant.

## 4. Discussion

The present study showed that functional constipation was associated with anorexia in community-dwelling older adults. Moreover, depressed mood and chewing problems are crucial confounding factors in their relationship.

Constipation is the most common chronic gastrointestinal disorder in adults and a frequent problem in the elderly [9,10]. The prevalence of constipation increases with age, up to approximately 50% and 60%, respectively, among adults over 80 years old and those in institutions [10,21,22]. In our study of community-dwelling older adults, 19.6% of the study subjects had functional constipation. In a previous study of 1954 adults in older nursing home residents, 1.1% of the subjects suffered from constipation [1]. According to another study conducted in acute and rehabilitation wards in Italy on a population of 98 adults, 59.8% of the subjects had functional constipation [2]. In an Italian study, anorexia was diagnosed as a decrease in food intake for more than three days without oral disorders, while constipation was diagnosed based on the weekly frequency of bowel movement. This discrepancy in the prevalence of anorexia and constipation may be due to differences in the definitions and evaluation methods used.

Anorexia is a particularly important geriatric syndrome because it accounts for the largest proportion of malnutrition in older adults. Many risk factors associated with this syndrome have been established, such as physiologic and pathologic functional impairment, social and environmental changes, acute and chronic diseases, and treatments due to aging [5].

Constipation has been suggested to be linked with anorexia of aging. Given that the prevalence of constipation increases rapidly with age, pathological mechanisms may overlap between anorexia in aging and functional constipation in older adults [23]. However, previous studies on the relationship between anorexia of aging and constipation showed conflicting results, and there has been no study about this relationship in community-dwelling older adults. To the best of our knowledge, this is the first study illustrating the association between anorexia and constipation in community-dwelling older adults. Both conditions may exist together due to the adverse effects of certain medications [8]. Pathologic mechanisms can also overlap between anorexia of aging and functional constipation in the elderly [8]. Changes in taste and olfactory functions, diminished hunger and altered satiety control mechanism (decreased serum ghrelin and neuropeptide Y (NPY) levels, and increased serum cholecystokinin (CCK), peptide YY (PYY), leptin, and insulin levels), and age-related gastrointestinal motility changes (decreased stomach compliance and delayed gastric emptying) are possible major physiological factors of anorexia of aging [5]. Age-related changes inevitably affect the lower gastrointestinal tract and the anorectal and pelvic floor muscles [23]. Pelvic floor dysfunction and anorectal disorders, which result in outlet dysfunction and the inability to adequately evacuate rectal contents, are also major changes that lead to functional constipation and anorexia in older adults [23]. Aside from these physiological changes, low food intake due to anorexia is a possible deteriorating factor for constipation in older adults [24]. Comorbidities can also structurally and functionally affect constipation and anorexia [24]. Therefore, functional constipation and anorexia may closely interact, resulting in a vicious cycle.

It should be noted that in this study, depressed mood and chewing problems appear to be confounding factors in the relationship between functional constipation and anorexia of aging. Depression is one of the most common psychological disorders among older people, and is often associated with the loss of motivation to eat in both community and institutional settings, which is one of the causes of anorexia of aging [2]. Among those diagnosed with depression, older adults seem to suffer more severe appetite loss than younger individuals [7]. The relationship between depression and constipation has been documented in two large population-based studies. In Iran, depression was found to be a significant predictor of chronic constipation [25]. Another study, using data from the National Health and Nutrition Examination Survey (NHANES) in the United States, found a higher proportion of depressed subjects with chronic constipation than non-depressed subjects [26]. Previous studies have proposed that mood may affect bowel movements due to an interaction between the central nervous system and the gastrointestinal tract, known as the brain–gut axis, and that depression increases intestinal transit [27]. According to other studies, chewing is significantly impaired in anorexic subjects [7]. Consequently, low caloric intake and a lower-fiber diet associated with a decline in food intake in the elderly may make a major contribution to constipation [26].

Polypharmacy is a strong risk factor for loss of appetite, and ultimately reduces food intake [28]. It is also known to be a risk factor for functional constipation in the elderly [20]. The risk of drug-induced anorexia is further increased by polypharmacy because of the enhanced odds of drug–drug interactions and gastrointestinal problems. However, in this study, polypharmacy was not a confounding factor in the relationship between functional constipation and anorexia during aging.

Loss of appetite and reduced food intake are also frequently observed in older adults with cognitive impairment, especially in later stages of the condition [4]. In this study, subjects with dementia were not included in the baseline enrollment in 2016. Therefore, there was a very low prevalence of dementia in 2018. Interestingly, in this study, subtle cognitive dysfunction was closely associated with anorexia of aging, but cognitive function was not an explanatory factor in the relationship between functional constipation and anorexia of aging.

This study has some limitations. First, the cross-sectional design of this study does not allow inference of causality in the relationship between functional constipation and anorexia of aging. Second, participants from our study were ambulatory community-dwelling older adults recruited using quota sampling methods. Therefore, our results may not be generalizable to other settings and populations. One key strength of this study, however, is its use of a nationwide sample of community-dwelling older adults. We also investigated the association of functional constipation assessed by the Rome IV criteria and anorexia of aging based on the total SNAQ score.

## 5. Conclusions

Our study showed that functional constipation is associated with anorexia in community-dwelling older adults. However, functional constipation was not an independent risk factor for anorexia of aging. Depressed mood and chewing problems were particularly important confounding factors in their relationship.

## Figures and Tables

**Figure 1 ijerph-18-05754-f001:**
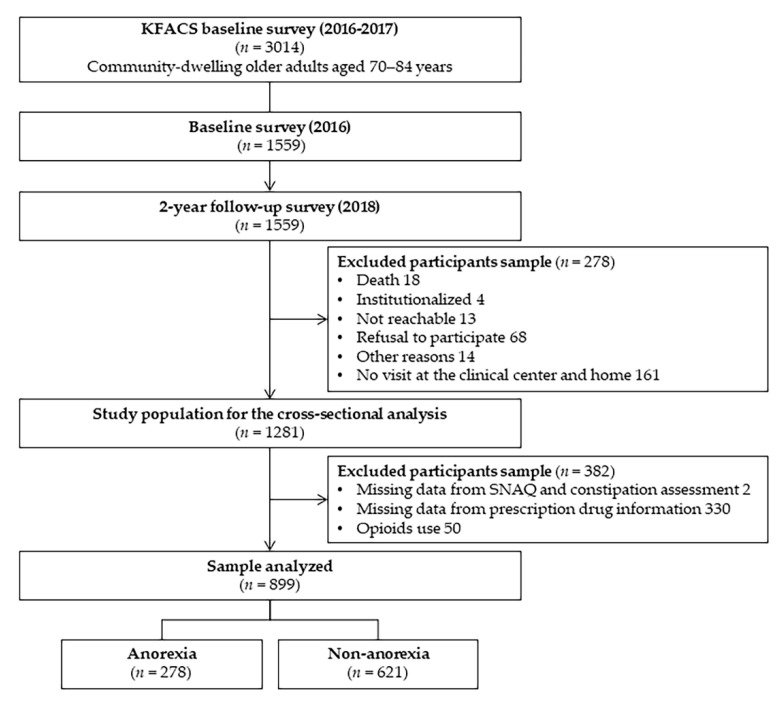
Flow chart of the study population.

**Table 1 ijerph-18-05754-t001:** General characteristics of community-dwelling older adults according to anorexia.

Variable	Anorexia SNAQ ≤ 13 (*n* = 278)	Non-Anorexia SNAQ > 13 (*n* = 621)	*p*-Value
**Demographics**			
Age	78.7 ± 4.1	77.9 ± 3.9	0.005
Female sex	177 (63.7)	285 (45.9)	<0.001
Living alone	80 (28.8)	151 (24.3)	0.157
**Clinical Characteristics**			
Health behavior			
Smoking	17 (18.7)	28 (10.2)	0.033
Alcohol drinking	54 (30.5)	181 (38.8)	0.050
Physical activity (MET-min per week)	392.5 ± 482.5	439.2 ± 527.0	0.208
Malnutrition (MNA ≤ 11)	87 (31.5)	76 (12.3)	<0.001
Clinical conditions			
HTN	169 (60.8)	402 (64.9)	0.232
DM	69 (24.8)	148 (23.8)	0.800
Dyslipidemia	118 (43.2)	272 (44.4)	0.735
IHD	7 (2.3)	14 (2.6)	0.813
Dementia	3 (1.1)	4 (0.6)	0.494
CVA	15 (5.4)	27 (4.3)	0.491
Depression (GDS ≥ 6)	112 (40.3)	100 (16.1)	<0.001
Thyroid disease	13 (4.7)	21 (3.4)	0.349
Kidney disease	1 (0.4)	12 (1.9)	0.070
Liver disease	2 (0.7)	4 (0.3)	0.406
Malignancy, remitted	11 (4.0)	19 (3.1)	0.489
Functional constipation	71 (25.5)	105 (16.9)	0.003
Chewing problems	149 (53.6)	189 (30.4)	<0.001
Polypharmacy	130 (46.8)	223 (36.0)	0.002
MMSE score (total)	24.65 ± 3.52	25.72 ± 3.33	<0.001
BMI	24.31 ± 3.36	24.56 ± 2.97	0.268
Weight loss for the past year	−0.21 ± 3.00	0.046 ± 2.48	0.224
Laboratory findings			
Serum albumin level (g/L)	4.30 ± 0.58	4.38 ± 0.31	0.023
High-sensitivity C-reactive protein (mg/L)	1.29 ± 2.01	1.14 ± 1.76	0.286

Notes: Data are presented as mean ± SD or as numbers (percentages). BMI = body mass index; CVA = cerebrovascular disease; DM = diabetes mellitus; GDS = Geriatric Depression Scale; HTN = hypertension; IHD = ischemic heart disease; MET = metabolic equivalent of task; MMSE = Mini-Mental State Examination; MNA = Mini Nutritional Assessment; SNAQ = Simplified Nutritional Appetite Questionnaire.

**Table 2 ijerph-18-05754-t002:** Association between functional constipation and anorexia of aging.

Variable	Model 1	Model 2	Model 3	Model 4	Model 5	Model 6	Model 7
	OR (95% CI)						
Functional constipation	1.682 *	1.543 *	1.473 *	1.478 *	1.223	1.423	1.117
	(1.195–2.367)	(1.087–2.188)	(1.035–2.097)	(1.038–2.104))	(0.845–1.770)	(0.995–2.034)	(0.765–1.631)
Age		1.053	1.047 *	1.040 *	1.046 *	1.041 *	1.028
		(1.015–1.093)	(1.009–1.087)	(1.001–1.080)	(1.008–1.086)	(1.003–1.081)	(0.988–1.070)
Female sex		2.072	2.078 *	1.934 *	1.878 *	2.059 *	1.840 *
		(1.544–2.781)	(1.547–2.792)	(1.435–2.606)	(1.388–2.541)	(1.525–2.779)	(1.348–2.511)
Polypharmacy			1.443 *				1.267
			(1.035–2.097)				(0.926–1.733)
MMSE				0.937 *			0.981
				(0.898–0.978)			(0.936–1.027)
Depressed mood (GDS ≥ 6)					3.052 *		2.568 *
					(2.186–4.260)		(1.801–3.661)
Chewing problems						2.464 *	2.196 *
						(1.827–3.322)	(1.612–2.992)

Notes: * *p* < 0.05 (statistically significant odds ratios). CI = confidence interval; OR = odds ratio; MMSE = Mini-Mental State Examination; GDS = Geriatric Depression Scale. Age: per year; MMSE: per 1 score; Polypharmacy: having five or more kinds of drugs per day. Model 1: Functional constipation; Model 2: Functional constipation adjusted by age and female sex; Model 3: Functional constipation adjusted by age, female sex, and polypharmacy; Model 4: Functional constipation adjusted by age, female sex, and MMSE score; Model 5: Functional constipation adjusted by age, female sex, and depressed mood (GDS ≥ 6); Model 6: Functional constipation adjusted by age, female sex, and chewing problems; Model 7: Functional constipation adjusted by age, female sex, polypharmacy, MMSE score, depressed mood, and chewing problems.

## Data Availability

The data presented in this study are available on request from the corresponding author. The data are not publicly available due to privacy reasons.

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
