# Peer review of "Functional Constipation and Anorexia in Community-Dwelling Older Adults: Korean Frailty and Aging Cohort Study (KFACS)"

_ijerph, 2021, doi:10.3390/ijerph18115754_

Round 1
Reviewer 1 Report
The introduction should be based a little more to place the context of the investigation and the state of the question The conclusions should indicate the specific contribution of this study compared to other research and improvement actions. Indicate the ethical issues in relation to the anomization of the participants and the ethical procedure or protocol (Declaration of Helsinki, etc.)
Author Response
Thank you for your comments. We have revised the manuscript in accordance with your recommendations.
Please see the attachment.

Reviewer 2 Report
The manuscript is properly designed and the study is scientifically sound. It is a longitudinal study involving many Centres. The tools that have been used to assess anorexia and cognition and bowel habits, are perfectly validated and appropriate. The statistical analyses are clear and easy to understand. I have no hesitation in accepting this study for publication.
Reviewer 3 Report
Thank you for this interesting work. I only have a few minor suggestions:
Line 75: Specify and explain the questionnaire mentioned.
Line 91: can you be more clear about what difficulty of giving opinion entailed and how it was measured?
Figure 1: add the date to the follow-up ïƒ 2018?
Line 99: should general be generally or in general?
Line 132: was there a questionnaire used to acquire the weekly activities that was then sed to determine METs?
Section 2.4: What were the methods of the clinical factors? For example the blood draws (fasting? Equipment?). Can you organize it as Sociodemographic factors then Clinical factors (so they are separate)?
Section 2.5: Were assumptions of the models checked?
Line 155: should specify the direction of the p value – example: statistical significance was determined as a p < 0.05.
Lines 159-161: can you add some of the data numbers here?
Author Response

(The authors gave the same response as above.)
